# Improving generalization in large language models by learning prefix subspaces

**Louis Falissard[1,2]**    **Vincent Guigue[3]**    **Laure Soulier[1]**

(1) Sorbonne Université, CNRS, ISIR, 75005 Paris, France
(2) Bibliothèque nationale de France, 75013 Paris, France
(3) AgroParisTech, UMR MIA-PS, 91120 Palaiseau, France
`louis.falissard@gmail.com, vincent.guigue@isir.upmc.fr`
`laure.soulier@isir.upmc.fr`

## Abstract

This article focuses on large language models (LLMs) fine-tuning in the scarce data regime (also known as the "few-shot" learning setting). We propose a method to increase the generalization capabilities of LLMs based on neural network subspaces. This optimization method, recently introduced in computer vision, aims to improve model generalization by identifying wider local optima through the joint optimization of an entire simplex of models in parameter space. Its adaptation to massive, pretrained transformers, however, poses some challenges. First, their considerable number of parameters makes it difficult to train several models jointly, and second, their deterministic parameter initialization schemes make them unfit for the subspace method as originally proposed. We show in this paper that "Parameter Efficient Fine-Tuning" (PEFT) methods, however, are perfectly compatible with this original approach, and propose to learn entire simplex of continuous prefixes. We test our method on a variant of the GLUE benchmark adapted to the few-shot learning setting, and show that both our contributions jointly lead to a gain in average performances compared to sota methods. The implementation can be found at the following link: `https://github.com/Liloulou/prefix_subspace`

## 1 Introduction

The emergence of large language models (Devlin et al., 2019; Radford et al., 2019; Raffel et al., 2019) in recent years has significantly transformed the applications of deep learning methods in natural language processing. These models, pretrained in an unsupervised fashion on massive textual datasets, enable the fine-tuning of powerful models with just a few thousand -or even hundred- observations. They achieve generalization performances that required millions of observations just a few years ago, particularly when used in conjunction with discrete instruction prompts (Brown et al., 2020).

Extending these discrete methods to the learning of continuous prompts (Lester et al., 2021), which conceptually falls within the framework of so-called "Parameter Efficient Fine-Tuning" (PEFT) methods (Houlsby et al., 2019; Bapna and Firat, 2019), poses certain challenges in the context of few-shot learning. One such challenge is the issue of model adjustment guidance through validation metric during gradient descent (Mao et al., 2022). Traditionally, in the process of model fitting, approximately one-third of the training dataset is excluded beforehand to create a validation (or development) set dedicated to inferring an unbiased estimation of the model's performance (Hastie et al., 2001). This metric is utilized both during gradient descent (to estimate convergence of the descent algorithm or inform early stopping heuristics), and subsequently to guide hyperparameter searches typically employed in the fine-tuning of large language models. However, the validity of this approach relies on the assumption that the distribution of the validation set is representative of the real observed phenomenon. This assumption quickly loses its relevance in the context of few-shot learning, where at most a few tens of observations are available for estimating the validation metric. This notion has become problematic enough in present times that a portion of academic literature on continuous learning with small datasets presents experiment results utilizing validation sets that are unrealistic and artificial, containing several orders of magnitude more observations than the training set used for the model adjustment itself (Mao et al., 2022).

In the machine learning community, characterizing local minima with desirable generalization properties has been a topic of interest for decades (Garipov et al., 2018; Zhang et al., 2021; Hochreiter and Schmidhuber, 1997). From flat minima (Hochreiter and Schmidhuber, 1997) to mode connectivity (Garipov et al., 2018), this body of work

has provided the basis for several practical observations regarding the connection between the properties of local minima and a model's generalization abilities.

The concept of learning neural network subspaces (Wortsman et al., 2021) is an example of a method built using these considerations. This approach proposes to find not just a local minimum of the cost function in the model's parameter space, but an entire simplex associated with low values of this objective. This additional constraint is meant to bias the descent algorithm towards wider minima, empirically associated with better generalization (Dziugaite and Roy, 2018a). In addition, the availability of this entire simplex of models allows for the inference of not only one scalar development metric, but an entire distribution, at any given moment during model fine-tuning. These two phenomena, become particularly relevant when viewed through the lens of large language models, and most especially for few-shot learning problems, where the model's ability to generalize a concept class from a limited number of examples is crucial.

The contributions of this article are as follows. First, we introduce the first adaptation of the subspace method to large language models through subspace adjustment of prefixes (a PEFT method similar to the state-of-the-art continuous prompt adjustment in current academic literature). Next, this article proposes to leverage certain natural advantages offered by the subspace method to revisit the concept of guiding model adjustment through the validation metric. We will empirically demonstrate that the combination of these two ideas leads to a significant improvement in terms of average prediction on natural language understanding tasks provided by the GLUE benchmark (Wang et al., 2018). Finally, an ablation study will be presented to provide some insights into the mechanisms underlying this prediction improvement.

## 2   Background

In this section, we review the two main concepts used in this article, neural network subspaces (Wortsman et al., 2021) and prefix-tuning (Li and Liang, 2021).

### 2.1   Mode connectivity and network subspaces

**Learning neural network subspaces.**   The subspace method proposes to obtain the simplex of solutions (in the parameter space of the studied model) through a single optimization loop as follows:

- A simplex of $n$ models is built through random initialization of each of its vertices using standard random initialization schemes.

- For each gradient descent iteration, a model is built as a weighted average of all vertices (to sample uniformly from the simplex they define)

- The sampled model is used for inference and cost function computation

- The gradient is backpropagated through all vertices to update all of their parameters

So far, the sampling procedure does not depend at all on the connectionist aspect of neural networks and simply considers a model as a vector of learnable parameters. However, the vast majority of deep learning models are defined as sequences of non-linear transformations (Goodfellow et al., 2016). Therefore, it seems natural to incorporate, in one way or another, this sequential structure of neural models into the sampling procedure. To do so, Wortsman et al. 2021 propose to sample each layer's parameters independently. This variant, known as the "layer-by-layer" method, is empirically associated with better generalization performances.

After model fitting, the simplex can be used either in the context of ensemble methods, or simply by using the simplex's centroid as the final model (Wortsman et al., 2021). The latter case is the one we focus on in this article, mainly because of the generalization properties it empirically displays.

**Subspace centroid and generalization.**   Several explanations have been proposed to explain these interesting generalization properties. One possible justification for this property, visualized in 1, lies in the idea that a model obtained through traditional training would be located at the periphery of a local minimum of the objective function, typically more susceptible to generalization errors (Izmailov et al., 2018; Dziugaite and Roy, 2018b). On the contrary, moving within the subspace allows us to "cross" the local minimum, in order to obtain a model associated with a more stable region of the objective function (Dziugaite and Roy, 2018a).

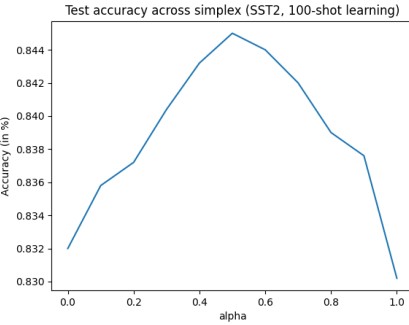

Figure 1: Generalization evolution of a language model adjusted on a prefix line, with alpha the weighting between both line extremities. Generalization performance follows a curve similar to a parabola with a maximum at its center.

**Application to LLMs** From the definition of the subspace method, it becomes clear why it has never been applied (at least to our knowledge) to large language models. First, it requires storing not just one model during the optimization loop, but all the vertices of the studied simplex. This additional memory complexity constraint is likely manageable in the case of adjusting a small convolutional network in computer vision. However, language models are known for their substantial size, reaching up to hundreds of billions of parameters, to the extent that the traditional fine-tuning of a single model already poses a considerable technical challenge for most specialized computing infrastructures. Therefore, the idea of simultaneously adjusting not just one but up to six models (the typical number of simplex vertices used in the subspace method) appears to be impractical.

In addition, and more fundamentally, this approach relies on a *random* initialization of the descent algorithm to construct an initial simplex of models. In contrast, pretrained language models are inherently initialized in a *deterministic* manner. Their entire transfer learning capabilities rely on the data representations captured into their parameter vectors during pretraining.

## 2.2 Prefix-tuning

On the other hand, continuous prompt adjustment methods and, by extension, PEFT methods (Houlsby et al., 2019; Hu et al., 2021; Li and Liang, 2021; Liu et al., 2022), propose not to directly fine-tune language models, but instead introduce new learnable parameters (such as the embeddings of virtual tokens in continuous prompt learning) and

adjust them while keeping the language model's pretrained parameters frozen. The main advantage of this approach lies in the ability of these "adapted" models to replicate (or even improve in contexts associated with small sample sizes) the performances of language models while reducing the number of learnable parameters by several orders of magnitude. In addition, some of these approaches (Li and Liang, 2021) typically require random initialization of the additional parameters they introduce into the model, making them particularly promising candidates for adapting the subspace method to large language models.

Prompt-based approaches (Liu et al., 2022), on the other hand, are based on the adjustment of $n$ embedding vectors $\{E_i\} \, i = 1^n$, typically concatenated at the beginning of the input embedding sequence of the language model $LM_\Phi$ parameterized by $\Phi$. In other words, for an input sequence of $L$ tokens, $\{I_i\}_{i=1}^l$, we construct a predictive model based not on the output of the language model itself:

$$LM_\Phi(\{I_i\}_{i=1}^l) \tag{1}$$

but on

$$LM_\Phi(concat(\{E_i\}_{i=1}^n, \{I_i\}_{i=1}^l)) \tag{2}$$

The adjustment of the predictive model is done solely by adjusting the virtual tokens $(E_i)_{i=1}^n$, while keeping the parameters of the language model $\Phi$ frozen.

To increase the expressiveness of this approach (which is particularly limited in terms of the number of learnable parameters), prefix tuning (Li and Liang, 2021), chosen in this article as a candidate for applying the subspace method, proposes to concatenate these virtual tokens not to the input sequence of the model, but to the Key and Value sequences used as inputs to the multiplicative attention modules in each layer of the language model.

In a similar approach to continuous prompt fine-tuning, the adjustment of prefixes is done solely by adjusting (via gradient descent) the virtual tokens, while keeping the parameters of the language model itself frozen. However, directly learning these embeddings proves to be particularly unstable (Li and Liang, 2021). Therefore, it is customary not to adjust them directly, but instead to use a reparameterization trick, which involves concatenating transformed versions of the prefixes to the Key and

Value sequences. This transformation is parameterized by a two-layer feed-forward network, as follows:

$$P_v = MLP_v(E) \ and \ P_k = MLP_k(E) \quad (3)$$

With :

- $P_v$ and $P_k$ the prefixes prepended to the Values and Keys sequences in the model, respectively

- $E$ a sequence of embedding vectors

- $MLP_v$ and $MLP_k$ the reparametrization perceptrons for the Values and Keys prefixes, respectively

## 3 Learning prefix subspaces

From this definition, it appears that the two main issues that made applying the subspace method to large language models cumbersome are both alleviated when using prefix tuning. Indeed, as a PEFT method, the number of trainable in the prefixes is orders of magnitude lower than the model's parameters, allowing to easily store simplexes in memory. Moreover, these prefixes are basically embedding vectors that require random initialization before model fine-tuning, which allows us to easily sample an initial simplex, by randomly initializing all its vertices.

### 3.1 Model formalization

The adaptation of the subspace method to prefix-tuning can be done through two distinct approaches:

1. Application to the learnable parameters of the model itself, namely the initial embedding and the reparameterization perceptron.

2. Application of the method to the prefixes themselves, specifically the output of the reparameterization module.

In this article, propose to investigate the second option, essentially considering the reparameterization module as a training artifact, and build our proposed prefixes as follows:

$$P_v = \Sigma_{i=1}^n \alpha_i MLP_{v,i}(E_i) \quad (4)$$

$$P_k = \Sigma_{i=1}^n \alpha_i MLP_{k,i}(E_i) \quad (5)$$

$$\{[\alpha]_1^n \in [0,1]^n ; \Sigma_i \alpha_i = 1\} \quad (6)$$

With :

- $P_v$ and $P_k$ the prefixes prepended to the Values and Keys sequences in the model, respectively

- $E_i$ the sequence of embedding vectors associated with simplex vertex $i$

- $MLP_{v,i}$ and $MLP_{k,i}$ the reparametrization perceptrons associated with simplex vertex $i$ for the Values and Keys prefixes, respectively

It is also important to consider the adaptation of the method's "layer-wise" variant. Indeed, the prefix adjustment does not rely on introducing a conventional perceptron structure into the language model, but rather on modifying the operation of the multi-head attention module. In this article, we propose to extend this "layer-wise" variant to each layer's Keys and Values prefixes. Thus, during each descent iteration, the Keys and Values prefixes of each layer will be independently sampled. Moreover, this sampling will be performed independently for all observations, unlike the traditional approach that prefers creating a single model per descent iteration step.

Additionally, the prediction head of the model is typically randomly initialized. Therefore, we choose to apply the subspace method to the prediction head as well, as described in Part 2. For consistency, the variant of parameter sampling at the observation level, rather than the batch level, will also be applied to it.

In summary, we propose to adjust a simplex with $n$ prefix vertices as follows:

- Independent initialization of $n$ reparameterization systems

- Computation of the $n$ vertices of the simplex for each descent iteration

- Construction of prefixes used for cost function inference and gradient calculation through independent uniform sampling for each observation, each layer, as well as for the prefixes of the Key and Value sequences.

### 3.2 Subspace learning and stochastic inference of development metrics

The adjustment of a large language model is typically guided by estimating a performance metric on a validation set, both during hyperparameter search and the descent process itself, where the best model according to this scalar value is selected as

the final model. The estimation of this metric in a subspace learning framework raises questions. Indeed, adjusting not just a single model, but an entire simplex, results in potentially estimable validation metrics.

Since we limit ourselves in this article to using this method to extract the centroid associated with better generalization performance, it would be natural to estimate the metric with respect to said centroid. However, the existence of not just a single model but this simplex, and the additional information it provides about the nature of the obtained local minimum, might be interesting. This is particularly the case in a few-shot learning context. As mentioned earlier, for validation datasets with small sample sizes (typically <100), estimating this metric can become unreasonably noisy.

Therefore, we propose using the entire simplex to "augment" the development metric's estimation. This will be done by not using the simplex' centroid for inference but by using *multiple* randomly sampled models for each observation from the validation set. In other words, for every development metric estimation, we propose to concatenate the development set multiple time, and to perform inference under the same conditions as during gradient descent iterations, meaning with randomly sampled models used for each observation. We set the number $n$ of developmlent set concatenation to 10 in all experiments presented in this article that employ this stochastic inference approach.

Nevertheless, we still select the centroid of the simplex as the final model. Indeed, the determinism of a model remains a desirable property in production settings.

## 4 Experimental protocol

### 4.1 Datasets

All experiments described in this article to evaluate the predictive performance of the proposed method are conducted with BERT-base-cased on datasets constructed from the GLUE benchmark (Wang et al., 2018), which consists of 8 English language comprehension tasks, all formulated as classification problems. However, these datasets have significantly larger sample sizes than what would be expected in the few-shot learning setting, and they do not make their test datasets available. As a consequence, we do not directly use these datasets, but instead adapt them to a format more

suitable to our problem using a methodology similar to that presented in (Mao et al., 2022). Namely, we build few-shot learning classification datasets with varying sample sizes (50, 100, 200, and 500 observations) through random sampling. However, our method of constructing these corpora differs from the original authors' on several key points.

First, the authors chose to construct validation sets of 1,000 observations for all their training datasets, which, in our opinion, is not realistic in a few-shot learning context (a validation set generally does not contain ten times more examples than its training set). Secondly, they use the GLUE benchmark's validation sets as test sets. However, some of these validation sets have small sample sizes (277 for RTE, 408 for MRPC), which could potentially introduce noise in the estimation of performance metrics. Therefore, for each reference dataset, a dataset of sample size $K$ is constructed as follows:

- The training and validation datasets are concatenated into a single dataset.

- Half of the observations (capped at 5,000 observations) are excluded from this dataset to construct a test dataset that is common to all experiments.

- $K$ observations are then selected through uniform sampling and divided into a training and validation dataset following a 70/30 proportion.

For each task in the benchmark and for each selected sample size, 10 datasets are constructed using this methodology to allow for replication of experiments on different datasets, enable estimation of average performance, and test the significance at a 5% threshold of the obtained differences (via bootstrap).

### 4.2 Baselines

All experiments described in this article to evaluate the predictive performance of the proposed method are conducted with BERT-base-cased. We compare our method to 5 baseline fine-tuning approaches, including standard fine-tuning and 4 alternate PEFT methods. Similar to prefix-tuning, these alternative fine-tuning approaches are based on the idea of freezing the language model's parameters and introducing a fraction of new adjustable parameters

(typically with a cardinality several orders of magnitude lower than that of the model itself), but they differ in how they introduce these new parameters into the model:

- Standard Adapter (Houlsby et al., 2019), which typically involves introducing one or more two-layer bottleneck perceptrons at different stages of a Transformer layer. This was the first PEFT method to be introduced and is the most recognized.

- Low-Rank Adaption (LoRA) (Hu et al., 2021), which reparametrizes the projection matrices of Values and Queries prior to the multi-head multiplicative attention module using two-layer linear bottleneck perceptrons. This was the first PEFT method to propose different transformations for different elements of the attention module.

- UniPELT (Mao et al., 2022), a fusion method combining adapters, LoRA, and prefixes to benefit from the advantages of each method (without suffering from their potential respective drawbacks).

- Standard prefix tuning, a crucial reference method to estimate the portion of the performance of the proposed method that can be attributed to it.

For the proposed approach and all aforementioned baselines, we follow the same procedure for model fitting and hyperparameter search as proposed by (Mao et al., 2022), and all experiment settings can be found in the annex. To ensure optimal comparability, the hyperparameter choices for the proposed method will be selected to exactly match those of the prefix tuning baseline, which were also determined for the first time in a text classification framework by (Mao et al., 2022). The subspaces adjusted in the experiments are all simplexes with 6 vertices.

## 4.3 Model variants

In order to better identify the impact of the different aspects of the proposed method, we also experiment with the following variants:

- Same method with 2-vertex simplexes (i.e., a line)

- Same method without stochastic validation inference

- Same method without prediction head subspaces

- Same method without prefix subspace (i.e., only on prediction heads)

## 5 Results

**Overall effectiveness** The performances of all selected PEFT methods, as well as the proposed approach, are presented in Table 1 for all different tasks of the GLUE benchmark and for the different selected sample sizes. Overall, the method significantly outperforms most baselines for all sample sizes, and notably outperforms significantly all baselines for $K = 500$. However, the method notably shows a higher gain for lower sample sizes ($K < 200$), which strongly implies that experiment results still suffer from high variance in this regime

The comparison between the proposed approach and prefix tuning is particularly interesting. Indeed, both approaches have the same exact functional form. In terms of statistical significance, the proposed method outperforms its classical counterpart 12 times:

- On QNLI, SST-2, and STS-B for K=50 and K=100

- On MRPC and QQP for K=200

- On MNLI, QNLI, MRPC, and STS-B for K=500

However, it is statistically surpassed only once, on MRPC for K=50, which is even more surprising considering that the difference between the two methods in this experiment is 0.4%. Moreover, the proposed method becomes significantly superior again on this task once the sample size increases up to 500 observations, showing a significant increase in generalization performances.

**Comparison between PEFT methods** More broadly, the proposed method is significantly surpassed only 6 times across all experiments:

- On MRPC by the prefix and LoRA methods for K=50

- On RTE by the Adapter method for K=50

- On STS-B by the UniPELT method for K=50

- On CoLA by the Adapter method for K=200 and K=500

| Method (number of params.) | MNLI | QNLI | SST-2 | QQP | CoLA | STS-B | MRPC | RTE | Avg. |
|---|---|---|---|---|---|---|---|---|---|
| *[K = 50]* | | | | | | | | | |
| Fine-tuning (108M) | 35.5 | 65.9 | 57.57$_*$ | 45.6$_*$ | **3.5** | 45.1$_*$ | 81.1 | 50.6 | 48.1$_*$ |
| Adapter (5M) | 35.6 | 62.6$_*$ | 64.7$_*$ | 35.3$_*$ | 0.0 | 59.7 | 80.2 | **53.1**$^*$ | 48.9$_*$ |
| LoRA (0.3M) | 35.7 | 63.9$_*$ | 68.4$_*$ | 47$_*$ | 1.0 | 56.5$_*$ | **81.4** | 52.8 | 50.8$_*$ |
| UniPELT (1.8M) | 35.3 | 62.4$_*$ | 73.1$_*$ | 42.3$_*$ | 1.1 | **64.3**$^*$ | 80.7$^*$ | 51.8 | 51.4$_*$ |
| Prefix-tuning (0.9M) | **37.8** | 63.5$_*$ | 74.9$_*$ | 53.1 | 1.8 | 59.2$_*$ | 80.4$^*$ | 52.6 | 52.9 |
| Prefix subspaces (0.9M) | 36.6 | **66.6** | **80.1** | **54.3** | 0.8 | 61.1 | 80.0 | 52.2 | **54.0** |
| *[K = 100]* | | | | | | | | | |
| Fine-tuning (108M) | 35.5$_*$ | 68.9$_*$ | 73.9$_*$ | 52.6$_*$ | 3.0$_*$ | 64.1$_*$ | 81.3 | 52.1 | 53.9$_*$ |
| Adapter (5M) | 36.3$_*$ | 66.7$_*$ | 72.8$_*$ | 54.0$_*$ | 7.2 | 63.8$_*$ | 80.5 | 53.0 | 54.3$_*$ |
| LoRA (0.3M) | 37.3 | 64.9$_*$ | 73.2$_*$ | 54.2$_*$ | 7.3 | 60.4$_*$ | 81.3 | 52.9 | 53.9$_*$ |
| UniPELT (1.8M) | 37.7 | 66.9$_*$ | 79.1$_*$ | 53.6$_*$ | 5.1 | 68.4 | 79.7$_*$ | 52.0$_*$ | 55.3$_*$ |
| Prefix-tuning (0.9M) | 38.3 | 69.4$_*$ | 80.8$_*$ | 57.2 | **8.1** | 66.6$_*$ | 81.1 | **54.2** | 57.0 |
| Prefix subspaces (0.9M) | **38.5** | **70.8** | **82.5** | **59.6** | 7.8 | 68.3 | **81.5** | 54.1 | **57.9** |
| *[K = 200]* | | | | | | | | | |
| Fine-tuning (108M) | 42.3 | **71.9** | 80.8$_*$ | 63.0 | 20.2 | 69.0$_*$ | 80.8 | 54.6 | 60.3$_*$ |
| Adapter (5M) | 42.7 | 69.1$_*$ | 83.1$_*$ | 59.5$_*$ | **26.5**$^*$ | 70.3$_*$ | 80.7 | **56.2** | 61.0 |
| LoRA (0.3M) | 41.0 | 67.1$_*$ | 82.2$_*$ | 61.2$_*$ | 19.8 | 67.8$_*$ | 80.1 | 54.5 | 59.2$_*$ |
| UniPELT (1.8M) | 41.6 | 70.2 | 82.8$_*$ | 58.7$_*$ | 16.4 | **72.8** | 81.7 | 54.9 | 59.9$_*$ |
| Prefix-tuning (0.9M) | **44.9** | 71.4 | **84.2** | 63.0$_*$ | 22.2 | 71.3 | 79.6$_*$ | **56.0** | 61.6 |
| Prefix subspaces (0.9M) | 44.7 | 71.2 | 84.1 | **64.4** | 21.1 | 72.3 | 81.6 | 55.9 | **61.9** |
| *[K = 500]* | | | | | | | | | |
| Fine-tuning (108M) | 52.7$_*$ | 74.3$_*$ | 85.4$_*$ | 66.8 | 32.2$_*$ | 78.0 | 82.5 | 59.8 | 66.5$_*$ |
| Adapter (5M) | 51.1$_*$ | 72.4$_*$ | 85.4$_*$ | 65.7$_*$ | **38.9**$^*$ | 76.1$_*$ | 81.9$_*$ | 59.8 | 66.4$_*$ |
| LoRA (0.3M) | 50.1$_*$ | 73.6$_*$ | 84.6$_*$ | 66.5 | 35.3 | 75.6$_*$ | 82.3$_*$ | 58.3$_*$ | 65.8$_*$ |
| UniPELT (1.8M) | 50.7$_*$ | 74.2$_*$ | 85.4$_*$ | 63.4$_*$ | 34.2 | 77.2 | 82.1 | 57.8$_*$ | 65.6$_*$ |
| Prefix-tuning (0.9M) | 54.0$_*$ | 74.7$_*$ | 85.6$_*$ | 66.2 | 35.7 | 77.8 | 82$_*$ | 60 | 67.0$_*$ |
| Prefix subspaces (0.9M) | **55.7** | **75.4** | **86.1** | **67.2** | 36.0 | **78.1** | **83.1** | **60.8** | **67.8** |

Table 1: Experiment results. F1 scores are reported for QQP and MRPC. Spearman correlation is reported for STS-B. Matthews correlation is reported for CoLA. Accuracy measures are reported for the remaining tasks. Results in bold and underlined correspond to the first and second best performances, respectively. Results followed by an asterisk as a subscript or superscript correspond to significantly higher or lower results compared to those of the proposed method, respectively.

It is noteworthy that most of these occurrences are observed for K=50 (and therefore validation sets of 15 observations), where model fitting becomes particularly challenging.

On the other hand, the proposed method significantly outperforms one of the other baseline methods in the conducted experiments a total of 80 times, demonstrating a clear advantage in terms of predictive power.

It is particularly noticeable that the majority of experiments where the proposed method outperforms the reference methods are mainly on three datasets: QNLI, SST-2, and QQP. Furthermore, the ability of the proposed method to significantly surpass the reference methods on these tasks does not seem to depend on the sample size of the datasets.

However, it is difficult to identify what distinguishes these datasets from those where the proposed method remains comparable to the reference

methods. Both groups feature an equal number of similar tasks and imbalanced datasets.

**Comparison between approach variants** The results for the investigated variants, which are displayed in Table 2, can be summarized as follows:

1. The use of two-vertex simplexes shows slightly inferior performance compared to the proposed method for K=50, and similar performance thereafter.

2. The use of the validation-guided subspace method estimated deterministically collapses for K=50, K=100, and K=200 (cases where the performance is even lower than the classical prefix fitting method) and eventually becomes equivalent to the proposed method.

3. The use of prefix subspaces without a head

| Method | $K = 50$ | $K = 100$ | $K = 200$ | $K = 500$ |
|---|---|---|---|---|
| Proposed approach | 54.0 | 57.9 | 61.9 | 67.8 |
| Line subspace | 53.6 | 57.9 | 62.1 | 67.6 |
| Deterministic | $49.5_*$ | 56.0 | 61.4 | 67.8 |
| Head subspace | 53.5 | 56.8 | 61.3 | 67.2 |
| Prefix only subspaces | 52.6 | 57.7 | 62.2 | 67.6 |

Table 2: Results of the ablation study. The reported scores correspond to the average predictive performances across all tasks in the GLUE benchmark.

subspace is consistently surpassed by the proposed method.

4. The use of prediction head subspace coupled with classical prefixes is considerably surpassed for K=50 (which is the only statistically significant result) and similar to the proposed method when the sample size increases.

These observations, taken as a whole, provide several pieces of evidence regarding the relevance of using the notion of stochastic validation metric inference in few-shot learning. In particular, Observation 3 shows that adjusting the prefix subspace with classical validation metric estimation is associated with performance gains similar to the proposed method only from $K = 500$ onwards. Moreover, the fact that the performance of this variant is even lower than that achieved by classical prefix adjustment further supports the importance of the proposed method of stochastic validation metric inference.

Subsequently, Observations 1, 2, and 3 provide slightly weaker arguments regarding the importance of simplex size in the context of this stochastic estimation. Although the simplex size does not appear to have an effect for $K > 50$ (strongly indicating that learning lines is preferable for these sample sizes, which are significantly more memory-efficient), it seems to have an impact for very small sample sizes. This observation could be explained by the richness of information extracted from the validation dataset through stochastic estimation, due to a larger simplex. However, the results presented in this article are insufficient to confirm or refute this hypothesis. Similarly, Observations 2 and 3 particularly highlight the importance of adjusting the entire set of learnable parameters of the model through the subspace when $K = 50$. This could also be explained by suggesting that restricting the stochastic validation metric estimation to a subset of learnable parameters limits the ability to characterize the obtained local minimum.

## 6 Conclusion

In this article, we introduced two innovative ideas. The first one, an adaptation of the subspace method for training large language models through the PEFT method, is, to our knowledge, the first example of its use in the academic literature on natural language processing. The second idea, proposing an alternative way to estimate development metrics, represents an original application of the subspace method and is not specific to problems encountered in textual data analysis. The combined use of these two methods leads to a significant improvement in the performance of common language models such as BERT on language comprehension tasks proposed by the GLUE benchmark, rephrased in a "few-shot learning" context. The ablation study presented at the end of the article also allows us to quantify the impact of these two contributions. The performance gains observed on very small datasets ($\leq 100$) seem to be mainly explained by the finer information extracted from the validation set through the stochastic metric estimation method. However, this gain appears to diminish for larger sample sizes, where the subspace method applied to prefix tuning seems to be sufficient on its own to achieve performance gains over standard PEFT methods as well as classical model training.

Finally, applying subspace learning to PEFT methods also enables the training of powerful predictive models while significantly reducing the computational resources typically required for training large language models. This approach preserves the fundamental efficiency goal of these methods. Learning prefix subspaces remains accessible even in situations where resources, both in terms of data and computational power, are limited.

# 7 Limitations

The proposed approach is meant to improve the performances of large language models in the context of few-shot learning. As a consequence, it becomes increasingly dependent on the type of representations the model's pretraining was able to capture. In other words, it would be highly unreasonable to expect the proposed method to perform well in highly complex tasks that cannot be easily captured by current unsupervised pretraining schemes.

In addition, the fact that this method allows for fine-tuning without high sample sizes in the development set might let people use it without any additional test set, and thus any model validation of any sort, which might lead to the implementation of highly biased models.

# 8 Ethical considerations

This article introduces a method for text classification that has the same exact functional form as a prefix fine-tuned large language model. As a consequence, they get the same exact ethical issues, such as socially biased classification algorithms. In addition, the method's increased generalization abilities make it so that these algorithms might be built with fewer observations, which can lead to ill-defined objectives. These concerns call for thought and caution when implementing tools using the proposed model.

# 9 Acknowledgements

We would like to thank the Sorbonne Center for Artificial Intelligence for funding Louis Falissard's post-doctoral contract within the MLIA laboratory of the Institute of Intelligent Systems and Robotics. We would also like to thank the ANR JCJC project SESAMS (Projet-ANR18-CE23-0001).

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

# A   Appendices

All models were trained for $50$ epochs using the default settings of the Huggingface Trainer, along with early stopping with a patience of 10 epochs. The batch size is fixed at $16$ for all experiments, and hyperparameter searches are performed through exhaustive search within the following values:

- Traditional fine-tuning: Learning rate from $[1e-5, 2e-5]$

- Standard adapter: Learning rate of $1e-4$ and reduction factor from $[3, 6, 12]$

- LoRA: Rank and alpha value fixed at 8, learning rate from $[1e-4, 5e-4]$

- UniPELT: Prefix length fixed at 10, adapter reduction factor fixed at 16, and LoRA with rank and alpha value fixed at 8. Learning rate from $[2e-4, 5e-4]$

- Prefix-tuning: Prefix length fixed at 50, learning rate from $[1e-4, 2e-4, 5e-4]$