# OpenReview forum: "Improving generalization in large langue model by learning prefix subspaces"
_EMNLP/2023/Conference — EMNLP 2023 Findings_

### Official Review · Reviewer_bmEc · 2023-07-31

**Typos Grammar Style And Presentation Improvements:** none.
**Soundness:** 3

**Excitement:**

2: Mediocre: This paper makes marginal contributions (vs non-contemporaneous work), so I would rather not see it in the conference.

**Missing References:**

none.

**Paper Topic And Main Contributions:**

This paper introduce the first adaptation of the subspace method to large language models through subspace adjustment of prefixes.

This paper propose to leverage certain natural advantages offered by the subspace method to revisit the concept of guiding model adjustment through the validation metric. And the author show that two ideas leads to improvement on natural language understanding tasks.

**Questions For The Authors:**

Please refer to **Reasons To Reject**. If the author can help understand their contribution clearly, I would be willing to raise my score.

**Reasons To Accept:**

The motivation is clear and reasonable.

The writing is clear.

**Reasons To Reject:**

Actually I am not familiar with this topic, but in my opinion, I think there exists some drawbacks:

 (1) The improvements is limited. From the Table 1, I notice that only when K is small (<100), the gain over the "Prefix-tuning" baseline is significant.

(2) Application scenarios are limited. As there are many types of tuning techniques (e.g. prefix/p-tuning/LoRA/..), the proposed method can only be applied on prefix methods.

(3) Due to the limited application, the experiments also seem to be  not enough.

**Reproducibility:**

4: Could mostly reproduce the results, but there may be some variation because of sample variance or minor variations in their interpretation of the protocol or method.

**Reviewer Confidence:**

1: Not my area, or paper was hard for me to understand. My evaluation is just an educated guess.

---

> ### Author Rebuttal · Authors · 2023-08-29
>
> [1 - Limited improvement]
>
> Thank you for your comments. We would like to point out that our proposed approach significantly outperforms the prefix-tuning baseline on 6 experiments with K>100, and moreover significantly outperforms prefix-tuning on average when K=500. We think that these experimental results strongly support the fact that our method offers some advantages over standard methods even for K>100.
>
> In addition, the case where K<=100 remains of particular interest to us since it is actually fairly common when designing real-world machine learning-based systems. We think that our experiments show the increase in stability and efficiency that our method yields in this specific setup.
>
>
> [2 - Limited application scenario]
>
> Although we only applied it to prefix-tuning, the subspace learning method can be applied directly to all the other PEFT baselines selected in our paper without loss of generality.
>
> First, P-tuning can be seen as a “simplified” version of prefix-tuning, where only one prefix is created, and concatenated to the model’s input, instead of having prefixes for each multi-head attention module. P-tuning prefixes are built using a reparameterization trick identical to the one used in prefix-tuning, making it directly compatible with our proposed approach.
>
> In addition, subspace learning was first derived for multilayer perceptrons and convolutional networks (Wortsman et al., 2021), and as such its application to Adapter or LoRA methods (both based on inserting small bottlenecked perceptrons in the pre-trained LLM) is straightforward.
>
> The choice of limiting ourselves to one PEFT method was made for the sake of conciseness, and to limit the computations in our experiments. Prefix-tuning was chosen as our candidate solely based on the fact that it was found to yield the best performances among all others in the scarce data regime.
>
> 1 - Mitchell Wortsman, Maxwell C Horton, Carlos Guestrin, Ali Farhadi, and Mohammad  Rastegari, 2021, Learning neural network subspaces
>
>
> [3 - Lack of experiments]
>
> We are sorry that you feel the experiments we displayed aren’t enough to demonstrate our method’s interest. We would like to emphasize the fact that PEFT methods are widely used techniques that have shown their interest in a broad range of applications, especially on very large models such as Llama, whose size renders traditional fine-tuning unreasonable.
>
> Our contribution, evaluated on prefix-tuning but potentially applicable to most (if not all) PEFT methods, constitutes a solution of choice for fine-tuning and stabilizing LLMs in the few-shot learning setting. We are convinced that this method has numerous potential applications and are currently working on some of them. We hope that the results we obtained so far have already demonstrated the value of this approach.

---

### Official Review · Reviewer_1Q35 · 2023-08-04

**Soundness:** 5

**Excitement:**

5: Transformative: This paper is likely to change its subfield or computational linguistics broadly. It should be considered for a best paper award. This paper changes the current understanding of some phenomenon, shows a widely held practice to be erroneous in someway, enables a promising direction of research for a (broad or narrow) topic, or creates an exciting new technique.

**Missing References:**

The hypothesis of a connection between flatter minima and better generalization in ANNs is not "recent," unlike what authors claim on L 074-079, and is generally traced at least to [2] and had substantial developments since, e.g. [3].

I believe at least those two citations are needed to make the theoretical foundation of the approach more clear and allow readers to better connect the proposed approach with existing body of knowledge, although a more in-depth introduction, both on flat minima and minima interconnectness would be welcome.

[2] Sepp Hochreiter and Jürgen Schmidhuber. Flatminima. Neural Computation, 9(1):1–42,1997.
[3] Chiyuan Zhang, Samy Bengio, Moritz Hardt, Benjamin Recht, and Oriol Vinyals. 2021. Understanding deep learning (still) requires rethinking generalization. Commun. ACM 64, 3 (March 2021), 107–115.

**Paper Topic And Main Contributions:**

The authors of the paper propose a novel method for few-examples fine-tuning of LLMs (few-shot learning, also referred to as Parameter-Efficient Fine-Tuning, PEFT). Their approach is based on a coordinated training of several models (model simplex / neural network subspaces) to improve the model's generalization. To go around the large parameter number of LLMs and the fact that the NN subspaces approach requires random initialization, the authors apply the model simplex approach to an existing approach in PEFT - prefix tuning, which trains a smaller model from randomization. This smaller model modifies the behavior of multihead attention modules, allowing a differentiable tuning of the LLM in a data-efficient way.

The authors refer to the resulting approach as the "Prefix Subspaces." They then demonstrate the performance of their approach compared to existing PEFT methods across a panel of tasks from the well-established GLUE benchmark, making sure to account for random fluctuation by 5-fold replication.

While not perfect, their method compares favorably to most existing methods. The authors then perform a performance regression analysis, showing that random initialization at the heart of the subspaces approach is critical for the performance of their method.

**Questions For The Authors:**

On line 434, the authors note that they used classical fine-tuning as a baseline comparison without specifying whether Adam or AdamW optimizer was used, given that the latter can be expected to perform better in low-resource fine-tuning [1]. In turn, that could modify the results presented in Table 1, although without affecting overall results or diminishing the authors' work importance and impact, which is the reason this remains a suggestion.

[1] Zhenxun Zhuang, Mingrui Liu, Ashok Cutkosky, and Francesco Orabona. Understanding AdamW through Proximal Methods and Scale-Freeness. arXiv:2202.00089.

**Reasons To Accept:**

The paper is exceptionally well-written and easy to follow. Including replicates and indicating statistical significance is a welcome addition, allowing the authors to better support their claims and demonstrate the advantages of their method.

The problem of data and resource-efficient transfer learning on LLMs is highly pressing, given the general public interest in fine-tuning pretrained LLMs to suit their specific applications better, and issues with the classical fine-tuning, namely the additional memory required for the optimizer and data batches, on top of storing non-quantized LLMs and the amount of data required for fine-tuning.

The 5-fold replication and associated statistical significance indication is a welcome improvement on the status quo in the algorithms research in ML and provides excellent support to author's conclusions and results.

To the best of my knowledge, the approach used by the authors is novel and is supported by the current understanding of interconnected and flat minima.

The authors' approach is elegant, and the evidence provided to support the authors' claims is convincing.

**Reasons To Reject:**

At this stage the only element missing for a complete accept is the lack of attached source code to be able to validate authors' approach and allow future work to build on top of it.

**Reproducibility:**

3: Could reproduce the results with some difficulty. The settings of parameters are underspecified or subjectively determined; the training/evaluation data are not widely available.

**Reviewer Confidence:**

4: Quite sure. I tried to check the important points carefully. It's unlikely, though conceivable, that I missed something that should affect my ratings.

**Typos Grammar Style And Presentation Improvements:**

- Regarding Table 1. While the information is comprehensive and is an appropriate amount of evidence for the authors' claims, I suggest that the authors add a set of heatmaps to convey the same information in a more rapidly understandable format.

- I would also suggest that authors provide a cut across the NNs in the simplex, showing the distribution of performance metrics, given their claim that taking into account all the models in the simplex rather than the mean was essential for the performance.

- L 553 - authors seem to refer to the ablation study as "investigated variants." I would suggest using a consistent naming scheme to make the reading of the paper easier.

---

> ### Author Rebuttal · Authors · 2023-08-29
>
> [1 - Reasons to reject]
>
> Thank you for your appreciation of our work and your comments.! The code source was uploaded to a GitHub that will be cited in the revised version of the paper upon acceptance.
>
> [2 - Question for authors]
>
> We apologize for this lack of precision. We indeed used AdamW, and not Adam, for all experiments in the article.
>
> [3 - Missing references]
>
> The two references you mentioned will be included in the paper, and the mention of mode connectivity being a recent notion will be altered to account for these works.
>
> [4 - Typos Grammar Style and Presentation improvement]
>
> Thank you for these ideas, we will include them in the revised version of our article.

---

### Official Review · Reviewer_vny6 · 2023-08-05

**Soundness:** 2

**Excitement:**

2: Mediocre: This paper makes marginal contributions (vs non-contemporaneous work), so I would rather not see it in the conference.

**Paper Topic And Main Contributions:**

The paper proposes PEFT which is a method for fine-tuning language models with subspace method can prefix tuning.

**Reasons To Accept:**

The proposed method obtains good results according to the experiment section in the paper.

**Reasons To Reject:**

It may not be appropriate to directly reject setting from previous works and only provide results of newly constructed evaluation dataset. Comparison with previous methods under their settings are still suggested for fair comparison.

Experiment on larger pre-trained language models are suggested.

Since the proposed method is claimed to be efficient, comparison of number of parameters are suggested to be added into Table 1.

The novelty is limited, the proposed framework is based on previous methods with little modification.

**Reproducibility:**

4: Could mostly reproduce the results, but there may be some variation because of sample variance or minor variations in their interpretation of the protocol or method.

**Reviewer Confidence:**

3: Pretty sure, but there's a chance I missed something. Although I have a good feel for this area in general, I did not carefully check the paper's details, e.g., the math, experimental design, or novelty.

---

> ### Author Rebuttal · Authors · 2023-08-29
>
> We would like to thank you for your review. Please, find below the responses to your different comments.
>
> [1 - Reject experimental setting from previous work]
>
> Our main issue with experimental settings displayed in most of the recent few-shot learning academic literature lies in the fact that they use validation datasets that are orders of magnitude larger than their corresponding training datasets. This is highly unrealistic, since the validation dataset is typically comprised of observations that are gathered from the same data generation process as the training dataset. As a consequence, we firmly believe that any insight gained from such an experimental setup would not be applicable to real-life scarce data regime problems.
> With this article, we also mean to establish a new standard in methodology for gradient-based few-shot learning. One that we feel is actually realistic, and could actually transfer to real-life few-shot learning situations. We could reproduce experiments with settings displayed in previous works if that were to change your evaluation of our paper. However, this would be at the expense of a considerable amount of computational cost (the experiments displayed in our paper required the fine-tuning of approximately 6 thousand models), which is why we chose not to do so in the original paper.
>
>
> [2 - Need for experiments on larger LLMs]
>
> Replicating the experiment on larger pre-trained LLMs would be highly desirable since they typically behave better in the scarce data regime. We have not done so yet due to computational and time limitations. Again the experimental setting displayed in our paper required us to fine-tune approximately 6K LLMs, which is highly demanding in terms of resources. We could not do these experiments in the amount of time available for the rebuttal. However, you are perfectly right, and investigating how our proposed approach translates to different LLMs (bigger, generative, with different pretraining) will be the object of future work.
>
>
> [3 - Missing number of trainable parameters in Table 1]
>
> Thank you for this remark, the number of trainable parameters will be included in Table 1. In the meantime, we will display it to you here:
>
> Standard fine-tuning: 108M
>
> Adapter: 5M
>
> LoRA: 300K
>
> UniPELT: 11M
>
> Prefix-tuning : 10M
>
> Prefix subspace: 10M (The final model derived from our method is functionally identical to one derived through standard prefix-tuning)
>
>
> [4 - Limited novelty]
>
> While it is true that PEFT and subspace learning are not new contributions of this article, we feel that another contribution of our paper, that is not mentioned in your summary of our article, lies in our proposal of using subspaces to derive distributions of validation metrics. To the best of our knowledge, this approach is truly novel and not limited to PEFT methods. It can indeed be applied to any neural architecture compatible with the subspace method (i.e. any perceptron, convolutional, or even transformer model), and its application to prefix-tuning is an experimental choice rather than a limitation. In addition, our experiments (especially our variants analysis) show their contribution to improving model performances, especially in the few-shot learning setting, which seems particularly significant to us.
> Finally, this paper is, to our knowledge, the first application of subspace learning to NLP and large language models. The contribution of subspace learning on LLMs, and of validation metrics, especially in the few-shot learning setting, seems particularly significant to us. To the best of our knowledge, we have found no trace of any existing work on the subject, and it is in addition key to avoiding the unrealistic validation sets currently used in the literature.

---

### Meta-Review · Area_Chair_gQms · 2023-09-18

**Recommendation:** 4

**Metareview:**

This paper combines the "Learning neural network subspaces" approach and the "Prefix tuning" to improve generalization of language models in few-shot learning setting. Experiments focus on K-shot (K=50,100,200,500) adaptation of Bert-base on GLUE benchmark. Reviewers find the existing evaluation sound, however recommend adding (a) experiments with full GLUE dataset (b) applying it with another PEFT method.  I recommend authors to consider adding these experiments as these experiments would help reader to understand limitations and potential of the main hypothesis of the paper better. These experiments should require << 6k experiments and worth the time.

---

### Decision · Program_Chairs · 2023-10-07

**Decision:**

Accept-Findings

**Comment:**

This paper combines the "Learning neural network subspaces" approach and the "Prefix tuning" to improve generalization of language models in few-shot learning setting. Experiments focus on K-shot (K=50,100,200,500) adaptation of Bert-base on GLUE benchmark. Reviewers find the existing evaluation sound, however recommend adding (a) experiments with full GLUE dataset (b) applying it with another PEFT method.  I recommend authors to consider adding these experiments as these experiments would help reader to understand limitations and potential of the main hypothesis of the paper better. These experiments should require << 6k experiments and worth the time.